Is coral richness related to community resistance to and recovery from disturbance?

Zhang Stacy Y. 1
Speare Kelly E. 1
Long Zachary T. 2
McKeever Kimberly A. 1
Gyoerkoe Megan 1
Ramus Aaron P. 1 2
Mohorn Zach 1
Akins Kelsey L. 1
Hambridge Sarah M. 1
Graham Nicholas A.J. 3
Nash Kirsty L. 3
Selig Elizabeth R. 4
Bruno John F. 1 jbruno@unc.edu
1 Department of Biology, The University of North Carolina at Chapel Hill , Chapel Hill, NC , USA
2 Department of Biology and Marine Biology, University of North Carolina at Wilmington , Wilmington, NC , USA
3 Australian Research Council Centre of Excellence for Coral Reef Studies, James Cook University , Townsville, QLD , Australia
4 The Betty and Gordon Moore Center for Ecosystem Science and Economics, Conservation International , Arlington, VA , USA
Johnson Magnus
Electronic publication date: 2014 Mar 18
Publication date: 2014
Volume: 2
Electronic Location ID: e308
Received 2013 Apr 25; Accepted 2014 Feb 25
Copyright: © 2014 Zhang et al.
Copyright year: 2014
Copyright holder: Zhang et al.
License: This is an open access article distributed under the terms of the Creative Commons Attribution License, which permits unrestricted use, distribution, and reproduction in any medium, provided the original author and source are credited.
License URL: https://creativecommons.org/licenses/by/3.0/

Keywords: Biodiveristy, Resilience, Stability, Coral reef, Disturbance, Recovery, Resistance, Community ecology

Funding: National Science Foundation NSF #1018291 #1017458 The University of North Carolina at Chapel Hill The Queensland Smart Futures Fund This research was supported by the National Science Foundation, the University of North Carolina at Chapel Hill, and the Queensland Smart Futures Fund. The funders had no role in study design, data collection and analysis, decision to publish, or preparation of the manuscript.

==============================
More diverse communities are thought to be more stable—the diversity–stability hypothesis—due to increased resistance to and recovery from disturbances. For example, high diversity can make the presence of resilient or fast growing species and key facilitations among species more likely. How natural, geographic biodiversity patterns and changes in biodiversity due to human activities mediate community-level disturbance dynamics is largely unknown, especially in diverse systems. For example, few studies have explored the role of diversity in tropical marine communities, especially at large scales. We tested the diversity–stability hypothesis by asking whether coral richness is related to resistance to and recovery from disturbances including storms, predator outbreaks, and coral bleaching on tropical coral reefs. We synthesized the results of 41 field studies conducted on 82 reefs, documenting changes in coral cover due to disturbance, across a global gradient of coral richness. Our results indicate that coral reefs in more species-rich regions were marginally less resistant to disturbance and did not recover more quickly. Coral community resistance was also highly dependent on pre-disturbance coral cover, probably due in part to the sensitivity of fast-growing and often dominant plating acroporid corals to disturbance. Our results suggest that coral communities in biodiverse regions, such as the western Pacific, may not be more resistant and resilient to natural and anthropogenic disturbances. Further analyses controlling for disturbance intensity and other drivers of coral loss and recovery could improve our understanding of the influence of diversity on community stability in coral reef ecosystems.

Introduction

A large body of recent and classical ecological research has focused on how biodiversity influences the stability of communities (May, 1973; McCann, 2000; Pimm, 1984; Hughes & Stachowicz, 2004; Steiner et al., 2006; Cardinale et al., 2012). Despite this interest, there is still no consensus on whether or why diversity influences stability. This may be due in part to the multitude of ways stability has been defined (Grimm & Wissel, 1997; McCann, 2000; Pimm, 1984). Here, we define community stability as the ability to maintain a given state regardless of perturbation, invasion, or extinction (McCann, 2000). Mechanistically, stability is made up of two components: resistance—the degree to which a community changes in response to a disturbance, and recovery or “resilience”—the rate of return to pre-disturbance conditions (Grimm & Wissel, 1997; Ives & Carpenter, 2007; May, 1973; McCann, 2000).

Species richness (the number of species within an area) is thought to influence community stability via several mechanisms (Ives, Cardinale & Snyder, 2005). For example, species rich communities are more likely, due to chance alone, to include greater functional diversity, and redundancy of function within functional groups—the insurance hypothesis (McNaughton, 1977; Tilman & Downing, 1994). The inclusion of species that can tolerate or benefit from disturbance could increase community resistance (McCann, 2000; Tilman & Downing, 1994; Yachi & Loreau, 1999). Additionally, positive feedbacks among species via facilitation may be more common in diverse communities, which may enhance post-disturbance recovery (Naeem, 1998; Naeem & Li, 1997; Yachi & Loreau, 1999).

Understanding the role of species richness in community stability has been a focus of a number of field studies and meta-analyses (Duffy et al., 2001; Ives & Carpenter, 2007; Stachowicz, Bruno & Duffy, 2007; Stachowicz et al., 2008; Cardinale et al., 2012). Some experimental studies in terrestrial plant communities have found a positive relationship between species richness and stability (Ives & Carpenter, 2007; Tilman & Downing, 1994). For example, more diverse plant communities appear to be more resistant to disturbances including drought (Tilman & Downing, 1994). Manipulations of marine plant and invertebrate community richness have found that diversity is positively related to resistance and recovery (Stachowicz, Bruno & Duffy, 2007). Likewise, Boyer, Kertesz & Bruno (2009) experimentally demonstrated that species-rich algal communities were more resistant to disturbances than species-poor communities in estuarine ecosystems of eastern North Carolina. One of the few large scale tests of the diversity–stability concept was a study of boreal forest communities in north-western Quebec (Grandpre & Bergeron, 1997), which similarly found that greater species-richness conferred more resistance to changes in community composition and abundance following a disturbance.

We tested the hypothesis that species richness is related to community stability (resistance and recovery) in coral reef ecosystems. Coral communities are affected by a large range of natural and anthropogenic disturbances including storms, disease outbreaks, coral bleaching due to ocean warming, dynamite fishing, and predator outbreaks (e.g., Acanthaster planci or Crown-of-Thorns Starfish (COTS)). A recent study of coral-community recovery found that disturbance type, e.g., “physical” vs. “biological” disturbances, and other reef characteristics had little effect on recovery rate; although, recovery was influenced by post-disturbance coral cover, geographic region, and reef management (Graham, Nash & Kool, 2011). We expanded on this work by asking whether coral richness is related to resistance to and or recovery from disturbances at a large spatio-temporal scale using coral cover as a metric of reef state. We hypothesized that coral communities in species rich regions would exhibit greater resistance and recovery in response to disturbances. We synthesized the results of 41 field studies documenting post-disturbance changes in coral cover across a natural gradient of coral richness.

Methods

Data sources and study selection

Our analyses were based on an expanded version of the coral cover change database compiled by Graham, Nash & Kool (2011). We added 18 new studies (35 additional reefs), resulting in a total of 82 reefs (Fig. S1). Twenty-eight studies were used in the decline analyses, and 30 were included in the recovery analyses, with many studies providing both resistance and recovery data. We identified and added studies that quantitatively documented community responses following disturbances, both losses and recoveries, in percentage of benthos covered by living coral species on subtidal reefs (0.5–40 m depth, median survey depth = 10 m). If component values were in a graphical format, ImageJ software was used to extract values for analysis. Survey locations were categorized into four regions: eastern Pacific, western Pacific, Indian Ocean, and Caribbean (Fig. S1). In addition to browsing relevant coral reef and ecology journals, we also used online academic search tools (e.g., Google Scholar, ISI Web of Science) to search for published peer-reviewed sources of coral-cover change data using the search terms: “disturbance”, “coral cover”, “Acanthaster planci,” “bleaching”, “storm”, “cyclone”, “hurricane”, “resistance”, “resilience”, “recovery”, and “richness”.

Studies had to meet four criteria to be included in the present analysis (Graham, Nash & Kool, 2011). (1) A clear description of sampling methods. Most surveys were performed using a variation of the line transect method where a 10–30 m transect is placed on the reef and coral cover is estimated either in situ or from digital images or videos. Although manta tow data can be unbiased if extensive training is involved (Miller & Müller, 1999), all manta tow data was excluded because we were unable to verify the accuracy of the majority of manta tow surveys performed. (2) Documented changes in coral cover due to one of three types of disturbance: Acanthaster planci outbreaks, coral bleaching, and storm (hurricanes and cyclones) impacts. Other disturbances such as dynamite fishing and ship groundings were not included due to a lack of studies and the small scale of disturbance. (3) Changes in hard coral cover were documented for resistance: pre-disturbance (initial cover) and immediately after disturbance (remnant cover); or recovery: remnant cover and peak recorded cover post-disturbance (peak recovery). Where peak recovery was assessed, post-disturbance cover was monitored for at least 3 years following disturbance. (4) Coral cover decreased at least 10% relative to pre-disturbance cover due to disturbance (Eq. (1)). (1) Initial cover−Post  disturbance  cover≥0.1Initial cover.

We excluded studies in which the change in coral cover was <10% relative to initial coral cover because such relatively small year to year changes cannot be reliably distinguished from random noise, e.g., stochastic variation and sampling error. Benthic survey methodologies rarely are able to detect changes in cover <5–10%. Graham, Nash & Kool (2011) used a similar justification for excluding such studies: “the initial drop in coral cover as a result of the disturbance exceeded 10%. Studies recording lower initial mortality were excluded as small disturbances may have negligible impact on hard coral cover (e.g., Edmunds, 2002). Furthermore, inter-annual variation in coral cover can be ∼5% prior to a disturbance (e.g., Halford et al., 2004). A 10% relative decline reflects a sudden change in benthic cover in response to a pulse disturbance”. However, in contrast to Graham, Nash & Kool (2011), we chose a 10% relative decline as a cutoff to account for disturbances that cause little absolute change but may in fact have a proportionally significant impact to reefs, particularly those with low levels of initial coral cover.

Predicted coral richness values in our analyses are geographic estimates based on coral species range distributions (Veron, 2000; Veron et al., 2011) and have been used by previous studies (e.g., Roberts et al., 2002; Veron et al., 2009). Maps were originally vector data, but we gridded them to a resolution of 50 km. The coral richness predictions that result from this database are relatively coarse-grained, however, given the strong association between local and regional coral richness (Karlson, Cornell & Hughes, 2004), our richness values likely correspond in a relative sense to local richness at a site.

Statistical analysis

We used the R programming environment (R Core Team, 2012) to perform two separate sets of regression analyses to test whether predicted coral richness was related to coral community resistance to and recovery from disturbance. Variance inflation factors (VIFs) were calculated for each pair of predictors to test for collinearity in both sets of analyses. The initial model for decline included predicted coral richness, initial (pre-disturbance) coral cover, maximum survey depth, region, and disturbance type as predictors. Initial predictors for recovery rate included predicted coral richness, post-disturbance coral cover, maximum survey depth, region, and disturbance type. We incrementally and individually removed non-significant predictors from the models using backwards stepwise elimination. Akaike Information Criterion (AIC) was used in model selection to assess which model provided the best fit. Decline (resistance) analyses were performed using square root transformed net absolute coral loss, and in the recovery analysis, we used log-transformed recovery rate, calculated as the change in coral cover, generally an increase, divided by the number of years until peak coral cover was observed (Eq. (2)). (2) logpeak recorded recovery−post-disturbance  coverpeak recovery year−post-disturbance year.

In cases where recovery was not observed and reefs continued to decline i.e., negative recovery, recovery rate was equated to zero. Transformations were based on the distribution of the raw data. All models were also performed on raw data, the results of which did not different qualitatively from those based on transformed data. The core database (including all response and predictor variables), the R code, and model outputs were made available for editors and reviewers upon manuscript resubmission and will remain available for future readers. We did this to enable our analysis to be reproduced, to be as open as possible about our procedures, and to facilitate future analysis based on our database.

Results

There was substantial variation among sites in predicted coral richness (24–552 species, Fig. 1), coral cover loss (Fig. 1), and recovery rate (Fig. 2). The full linear regression model testing the null hypothesis that coral species richness was not related to (sqrt transformed) post-disturbance coral cover loss (decline) found that all covariates (including richness; p = 0.79) except pre-disturbance coral cover (BLCC, p < 0.0001) were unrelated to decline (Fig. S2, Table 1, Model 1, Adj. R2 = 0.63, see all model results and R code in the Statistical Supplement). A simpler model (Model 2, Adj. R2 = 0.60) that included three covariates (depth, richness, and BLCC) with a similar fit indicated that the richness effect was marginally significant (p = 0.03). There was no relationship (Adj. R2 = 0.01, p = 0.50) between predicted coral richness and initial cover, thus it is unlikely that collinearity between these two covariates led to the putative effects of richness on resistance. When we removed sites (two potential outliers) with extreme decline and high diversity, the relationship between relative decline and predicted species richness was still significant (in the reduced model with three covariates).

Figure 1 Coral community richness and resistance to disturbance.

3-D scatterplot of the relationship between coral community richness, coral loss (net change in absolute percent cover in response to disturbance) and initial coral cover. See Table 1 for results of statistical analysis. Dashed lines represent the regression plane, and point coloration represents region. Refer to the Statistical Supplement for a version of this figure plotted with the response variable square root transformed.

Figure 2 Coral community richness and recovery from disturbance.

Scatterplot of the relationship between the post-disturbance recovery of absolute coral cover and estimated coral species richness. Refer to the Statistical Supplement for a version of this figure plotted with the response variable log transformed.

The full recovery model (Table 1, Model 4 in the Statistical Supplement, Adj. R2 = 0.27) was the best fit and indicated that none of the covariates were related to recovery rate. Predicted coral richness in particular appeared a very poor predictor of coral recovery rate (p = 0.56).

Table 1 Linear models performed in analyses.

Linear model description—decline	
Raw data	AIC	R2​	p-value	
Richness + maxdepth + disturbance + region + BLCC	351.94	0.71	4.04e−07	
Richness + maxdepth + BLCC	349.46	0.65	5.56e−09	
Richness + BLCC	414.92	0.60	2.73e−10	
Richness + maxdepth + disturbance + region + BLCC
+ maxdepth:disturbance interaction	354.75	0.71	2.85e−06	
Square root transformed				
Richness + maxdepth + disturbance + region + BLCCb	148.77	0.70	5.62e−07	
• Intercept			0.05	
• Richness			0.78	
• Max depth			0.95	
• Disturbance—COTS			0.69	
• Disturbance—Storms			0.51	
• Region—E. Pacific			0.17	
• Region—Indian Ocean			0.67	
• Region—W. Pacific			0.98	
• BLCC			3.72e−8	
Richness + maxdepth + BLCC a,b	147.60	0.63	1.50e−08	
• Intercept			0.15	
• Richness			0.03	
• Max depth			0.61	
• BLCC			1.02e−08	
Richness + BLCC	174.83	0.59	6.82e−10	
Richness + maxdepth + disturbance + region + BLCC
+ maxdepth:disturbance interaction	150.30	0.72	2.49e−06	
				
Linear model description—recovery	
Raw data	AIC	R2​	p-value	
Richness + maxdepth + disturbance + region + PDCC	264.05	0.27	0.03	
Richness + maxdepth	264.11	0.12	0.04	
Richness	274.63	0.062	0.07	
Log transformed				
Richness + maxdepth + disturbance + region + PDCCa	104.77	0.38	0.04e−1	
• Intercept			3.95e−06	
• Richness			0.56	
• Max depth			0.10	
• Disturbance—COTS			0.35	
• Disturbance—Storms			0.21	
• Region—E. Pacific			0.25	
• Region—Indian Ocean			0.46	
• Region—W. Pacific			0.46	
• PDCC			0.13	
Richness + maxdepth	109.21	0.15	0.01	
Richness	115.60	0.11	0.01	
Notes.

a Indicates best fit decline models which did differ significantly based upon AIC.

b Indicates overall best fit models for coral cover decline and recovery based upon lowest AIC.

p-values for best fit model factors listed below model description.

BLCC Baseline (prior to disturbance) coral cover

PDCC Immediately post-disturbance coral cover

Discussion

We did not detect an effect of region, disturbance type, or depth on absolute loss in coral cover due to disturbance (Table 1, Statistical Supplement). There was a weak positive relationship between predicted species richness and coral loss (Fig. 1). However, a richness effect (p = 0.03) was only detected in the reduced model (Model 2), from which maximum depth, region, and disturbance type were removed (Statistical Supplement). In the full model with all five covariates (Model 1), the richness effect was not significant (p = 0.79).

Concordant with several previous studies (Graham et al., 2008; Selig & Bruno, 2010; Selig, Casey & Bruno, 2012), reefs with greater initial cover lost the most coral following disturbance (Fig. 1, Statistical Supplement). This could be due in part to the dominance of high cover reefs in the western Pacific and Indian Ocean by plating acroporid corals, that are especially sensitive to both physical and biological disturbances (Darling et al., 2012), i.e., not due to a cover effect per se, but instead to the relationship, both among reefs and regions, between coral cover and species composition. Acroporids are among the most sensitive coral species to most of the disturbances known to cause declines in coral cover, e.g., thermal stress, disease, predation, and physical disturbance (Bruno et al., 2001; Darling et al., 2012; Willis, Page & Dinsdale, 2004). Therefore, reefs dominated by plating and branching acroporids–the highest cover reefs–are likely to be especially susceptible to disturbance. Although high taxonomic richness may confer greater functional diversity and redundancy, this also implies a greater number of specialists that may not be well adapted to disturbance or stressful conditions (Dobzhansky, 1950; Schluter, 2000) and may partially explain our results. In fact, Côté & Darling (2010) argued that managing for coral cover could counter intuitively make reefs more susceptible to disturbances including climate change.

Unlike several previous experimental studies in marine systems (Allison, 2004; Reusch et al., 2005; Williams, 2001), our mensurative study found that the rate of post-disturbance recovery was not significantly related to taxonomic richness (Table 1, Fig. 2). The large observed variance in recovery rate among sites and studies (up to 12.5% per year) appears to be due to factors not included in our model and unrelated to estimated species richness.

Graham, Nash & Kool (2011) reported that coral recovery varied somewhat among regions, with the highest observed recovery rates in the western Pacific, the most species rich region, (Roberts et al., 2002). Despite that finding, in our analysis coral richness was a poor predictor of recovery rate (Fig. 2); although, the low richness reefs of the eastern Pacific recovered slowly, recovery of Caribbean reefs with similar richness was greater and similar to that seen in the more diverse western Pacific and Indian ocean (Fig. 2). This finding could be due in part to a publication biases towards studies that document recovery and the fact that we searched for studies documenting acute disturbances; whereas, chronic perturbations are also responsible for Caribbean coral losses. Furthermore, studies conducted in the eastern Pacific and included in our analysis were mostly conducted on reefs dominated by slow-growing, massive Porites. Given the current state of the Caribbean (Bruno et al., 2009) there is no doubt that coral cover on most Caribbean reefs has not recovered from various natural and anthropogenic disturbances over the last several decades (Roff & Mumby, 2012). Hence, our results suggest that low-diversity Caribbean reefs have the potential to recover (in terms of coral cover) quite rapidly (e.g., Idjadi et al., 2006), but for a variety of reasons and factors outside the scope of our study, they rarely do (Roff & Mumby, 2012).

The majority of reefs in our study were monitored for less than eight years and few were monitored long enough to recover fully (Fig. S3), given observed rates of recovery. Of the 23 reefs for which we had both initial post-disturbance and peak-year coral cover, only five (all monitored for at least 11 years after disturbance) recovered to pre-disturbance levels of cover. If recovery (coral cover increase) is indeed non-linear over time, we could have seen different results, e.g., a significant richness effect if the component studies were conducted longer. To a degree, our results could also be an artifact of the loss of Acroporid corals from Caribbean reefs in the 1980s, due primarily to disease but in some locations to storms and predation (Aronson & Precht, 2006). Our earliest Caribbean recovery study began in 1987 (Edmunds, 2002), after Acroporid populations had crashed regionally (Aronson & Precht, 2006). Had Acropora cervicornis and A. palmata been present on Caribbean reefs when the datasets used in our analyses were collected, we might have seen more rapid recovery on low diversity reefs because these corals were amongst the fastest growing corals in the Caribbean region. However, we may also have seen even less resistance to disturbance due to their high coral cover and vulnerability to various stressors.

It is possible that systematic variation in disturbance intensity could obscure a richness effect on coral community resistance. For example, disturbance intensity may have co-varied with coral richness or peak coral cover. Although quantification of the intensity of disturbance is beyond the scope of this study, it would, at least in theory, be possible. Our study also addresses a very large scale richness gradient but ignores smaller scale variation, e.g., among reefs or zones (Huston, 1985). Furthermore, we used predicted richness values from known biogeographic ranges (Veron, 2000; Veron et al., 2011), which should reflect patterns of richness at this scale, but can make richness and region hard to tease apart and cannot account for local site-level differences due to factors such as reef zone, disturbance history, or micro-climates. Diversity of fish assemblages may also be critical for coral recovery dynamics (Bellwood et al., 2004), which was not considered in this study. Future analyses with empirical diversity data of corals and fish, controlling for disturbance intensity and other drivers of coral loss and recovery, could improve our understanding of the influence of diversity on community stability in coral reef ecosystems.

Supplemental Information

Figure S1 Map of reef locations from studies used in the meta-analysis

Also see the live version in Google Maps.

Click here for additional data file.

Figure S2 Coral resistance and recovery for different regions and disturbance types.

Comparisons of coral cover decline from the resistance analysis (A–C) and post-disturbance coral cover increase from the recovery analysis (D–F) among regions and disturbance types. Boxes are interquartile ranges and lines within each box represent median values. Error bars extend to the minimum and maximum values.

Click here for additional data file.

Figure S3 Frequency distribution of study duration for data sources used in the recovery analysis

Click here for additional data file.

Supplemental Information 4 Metadata for the file “reefresilience_data.xls”

Click here for additional data file.

Supplemental Information 5 Study data

Click here for additional data file.

Statistical Supplement Statistical supplement

Complete R code, model outputs, etc. for the primary analyses

Click here for additional data file.

Text S1 References for data sources

Click here for additional data file.

Additional Information and Declarations

Competing Interests

Author Contributions

Data Deposition

John F. Bruno is an Academic Editor for PeerJ. Elizabeth Selig is an employee of The Betty and Gordon Moore Center for Ecosystem Science and Economics.

Stacy Y. Zhang conceived and designed the experiments, performed the experiments, analyzed the data, contributed reagents/materials/analysis tools, wrote the paper, prepared figures and/or tables, reviewed drafts of the paper.

Kelly E. Speare, Kimberly A. McKeever, Megan Gyoerkoe, Aaron P. Ramus, Zach Mohorn, Kelsey L. Akins and Sarah M. Hambridge conceived and designed the experiments, performed the experiments, wrote the paper, reviewed drafts of the paper.

Zachary T. Long analyzed the data, contributed reagents/materials/analysis tools, wrote the paper, reviewed drafts of the paper.

Nicholas A.J. Graham and Kirsty L. Nash performed the experiments, contributed reagents/materials/analysis tools, wrote the paper, reviewed drafts of the paper.

Elizabeth R. Selig contributed reagents/materials/analysis tools, wrote the paper, reviewed drafts of the paper.

John F. Bruno conceived and designed the experiments, performed the experiments, analyzed the data, wrote the paper, reviewed drafts of the paper.

The following information was supplied regarding the deposition of related data:

FigShare:

Bruno, John (2014): Coral richness, resistance, and recovery data. figshare.

http://dx.doi.org/10.6084/m9.figshare.941062.

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
