# Peer review of "Is coral richness related to community resistance to and recovery from disturbance?"

_PeerJ, doi:10.7717/peerj.308_

## Round 0.1 · original submission · Major Revisions

Having reviewed both the paper and the reviewers comments, I concur that although it is very well written it requires a major review. I also agree with the reviewers indications that with a bit more work this could be an excellent and important paper. And I wholeheartedly agree with Reviewer 2 - whether you get this through the review process or not, you have done a wonderful thing with your students.

As it is a meta-analysis, if possible, I would like to see more of the standard checks used to investigate the potential effects of sample size and you may wish to consider weighting your model based on this. I find "Borenstein et al (2009) Introduction to Meta-analysis", Wiley very helpful. There are also some great packages available for meta-analysis in R, e.g. meta and metafor with some fairly easy to navigate support documentation.

Some minor comments:
Line 18: classical rather than classic?
Line 140: Darling and Cote (2010) is not cited in the reference list but Cote and Darling (2010). Ususally one error in a reference list is indicative of more - I suggest a careful check.
Line 148: remove "value from"
Line 150: Replace "Like" with "As with"?
Line 152: I find it hard to believe that Richness in Fig 2 is normally distributed. Given that there are two obvious clumps of data I'd quite like to see the evidence demonstrating normality.
Line 152: What are the different point characters in figure 2? A figure always benefits from having a legend that explains it in enough detail so that the reader can understand it without having to search through the main text.
Line 152: Would recovery be a better word than restoration?

·

Basic reporting

While I really liked the conceptual framework of Zhang et al.'s piece on the diversity and stability relationship in coral reef communities, I'm afraid that it needs some serious reworking of its analyses before it is ready for publication.

Experimental design

At its core, the piece is an observational study using a marvelous database of decline and recovery after disturbances on a variety of coral reefs across the globe. However, observational studies that wish to discuss causal relationships do not have the luxury of experiments in implementing strict control of conditions. Therefore, analyses need to be carefully controlled for which variables enter a model based on best knowledge of the biology of a system, and all data must be considered carefully. Unfortunately, Zhang et al.'s treatment of this dataset and their analytic framework has serious flaws on both these lines that would make a skeptical reader very wary of their results.

Validity of the findings

First, what describes a disturbance and data filtering? While the authors have a documented disturbance type for all data points in this analysis, they throw out any data where coral cover decreases by less than 10%. While on the surface one could argue, "well, that wasn't a disturbance, then", that argument conflates applied disturbance intensity with realized disturbance intensity. Yes, a storm hitting a reef may not remove much coral at one site. But at a nearby site, the same storm may remove a huge amount of coral. It is precisely the difference between these two reefs that is so interesting, and might serve as useful data for the analysis. Throwing out the <10% data thus presents a real problem, particularly when decline in coral cover is a response variable in the resistance to disturbance model later in the paper. This may mean considering generalized linear models instead of linear ones, but that's OK. It would add a fair dash of realism.

Second, the backwards elimination to remove depth, region, and disturbance type strikes me as data dredging. There is no a priori reason for dropping them. Indeed, a stringent test of the hypotheses presented here would want to control for these variables. If one or more of these variables was correlated with richness, its removal could lead to variation due to that variable being folded in to the diversity effect. This is particularly problematic if some of the signals are weak. The reader cannot evaluate this, however. Indeed, a Neyman-Pearson p<0.05 decision rule without reporting to the reader relevant p values keeps us from being able to judge whether there may have been some weak signal that needed to be controlled for in the final analyisis.

Third, and related, there are a wide variety of biotic and abiotic predictors - temperature, productivity, fetch, disturbance history, herbivorous fish abundance, pH, etc., etc., etc., that could all be equally valid explanations for the observed pattern and could also be highly correlated with diversity. Thus, any results here may, for example, merely reflect disturbance history - a highly disturbed reef is low in diversity, but also likely to be not so resilient to future disturbance. There is simply no way of knowing, and any reader with more than moderate skepticism will come up with a wide variety of scenarios that could produce a spurious correlation here. The answer to such thinking is not to throw up one's hands in frustration, but address the problem head on! With careful consideration of additional covariates as well as how factors which may be beyond the analysis are correlated with diversity (say, at a subset of reefs for which the measurement exists), the authors can control for spurious results, and also conduct a much more nuanced and useful discussion after evaluation of these more detailed and biologically realistic models.

Additional comments

In short, there are some great ideas here. I like how they relate to previous experimental work. However, a skeptical reader is highly likely to dismiss the results out of hand as lacking proper controls and accounting for past knowledge of the ecology of disturbance. If more careful and thoughtful analyses are conducted with this data, however, I think that some very interesting results could pop out.

-Jarrett Byrnes

·

Basic reporting

The paper is very well written. The introduction sets the stage well.

Experimental design

The goal is clear, as is the description of the methods. The analyses are based on simple backwards stepwise regressions. For the analysis of decline, it seems that four factors were initially included: depth, region, disturbance type and initial cover (coral species richness was also included but not mentioned in this part of the methods). For the analysis of recovery, it seems that only depth and richness were initially considered. It's not clear to me why the other factors were not also included in the recovery analysis. There should also be some consideration of biologically meaningful interactions, such as region x disturbance type. Fig S3 shows what seem to be some differences in responses that are variable across regions and disturbances.

The authors should also justify why they limited their selection of studies to those showing at least a 10% decline in coral cover. This means that this is a study of stability in the face of acute or point disturbance.

Validity of the findings

The main findings are that absolute decline in coral cover in response to an acute disturbance increases with initial coral cover and coral species richness (so diversity does not increase this aspect of stability), but the rate of coral cover recovery increases with coral species richness (so diversity increases this aspect of stability).

The lack of a region effect is puzzling. Initial coral cover seems to have a large effect on absolute loss (parameter estimates for the effect size of initial cover and species richness would be useful), and I would assume that initial cover would have been lower on Caribbean reefs, largely due to the absence of acroporids. I really would like to see a comparison of initial cover among regions, and perhaps also within-region analyses. The latter would control for a lot of potentially confounding factors (such as initial coral cover, the loss of fast-growing but susceptible acroporids in the Caribbean, and the regionally variable types of disturbance).

The authors state that quantifying the intensity of disturbances is beyond the scope of the study. In fact, they can capture something of the intensity of disturbances since it is reflected in the extent of absolute loss in coral cover. So it should be possible to include coral loss (as a proxy for disturbance intensity) as a potential factor explaining coral cover recovery for those studies reporting both loss and recovery.

A more troubling issue is the variation among regions in the frequency of the disturbances. Gardner et al. (2005, Ecology), for example, showed that frequently recurring hurricanes had increasingly small effects on coral loss (because everything that could be broken was broken by the first hurricane). If this pattern holds for other types of disturbance, and the effect is regionally variable, it could drive some of the patterns observed here.

Additional comments

This is an interesting paper, and I love the fact that undergraduates played a large role putting this paper together. What a great way to learn!

The MS got me wondering about the difference between stability and resilience. The definition of stability adopted here is essentially resilience (i.e., a two-part process: resistance and recovery), which seems appropriate when the focus is on a pulse disturbance. For many people though (e.g. many of the papers cited in the introduction), stability means something more long-term than change around an acute disturbance, i.e. stability is measured by variability (or lack thereof) over time. This seems to be closer to the idea of dynamic stability of McCann (2000). So the diversity-stability relationships tested here are not exactly comparable to previous efforts (see, for example, Fig 2C in Ives and Carpenter 2007, for an example of positive diversity-stability relationship in a system experiencing a pulse disturbance). It doesn't mean that they are invalid, but some discussion of how short-term trajectories of coral decline and recovery might relate to longer-term trajectories might be useful.

Reviewer 3 ·

Basic reporting

The manuscript is generally well-written and easy to follow.

I recommend re-visiting the literature cited to incorporate some key papers on this topic - e.g., Cardinale et al. 2012, Naeem et al. 2012 (and references within re: diversity and stability).

More specific notes re: references -
1. References within the text are not given in chronological order as is the standard convention.
2. Lines 38-39: these references are also appropriate in lines 19-20.
3. Line 41: Tilman 1996 is another good reference here.
4. Lines 151-152 - The Reusch and Williams references are for manipulations of genetic diversity, not species diversity. This distinction should be noted.

Minor additional comments -
1. In the abstract, the authors state that the effect of species richness on recovery is marginally significant - a more accurate statement would be that the relationship is weak or has little explanatory power (i.e., low R2) rather than marginal significance (P-value).
2. Line 25 - the rate of recovery is also often referred to as 'resilience'
2. What do the symbols in Figure 2 represent?

Experimental design

The statistical analyses section would benefit from greater detail. Items to address include:
1. How/why were the variables included in the preliminary analyses for decline (lines 107-108) selected?
2. Were these same variables initially included in the recovery analyses? If not, why not?
3. Is there a relationship between peak recovery and years of observation? The authors suggest that recovery is non-linear (lines 172-174), but it is not clear why.
4. Please provide more information on the stepwise procedure used to identify the best model. Was it simply stepwise regression? If so, what parameters were used? The authors may consider using a model selection approach with AIC as an alternative method of exploring the relationships among richness and resistance and recovery.

In addition to providing greater detail, the authors should consider (and address) the following shortcomings of their analyses:
1. Given the importance of post-disturbance cover and geographic region on recovery (lines 58-60), why were these variables not explicitly included in the recovery analyses conducted for this study?
2. Species richness in this study is confounded with geography in several important ways - for one, species richness and composition co-vary geographically (lines 158-160) and likely influence both resistance and recovery from disturbance, and two, the values of richness used are geographically-based at a fairly coarse scale. It would be interesting to see how the inclusion of some metric of geographic location would change the results presented here. Given that the authors interpret their analyses from a geographical perspective (lines 192-196), it seems critical to include this variable in the analyses. The methods suggest that geographic region was included in the resistance analyses - was it insignificant? And was it ever included in the recovery analyses?

Validity of the findings

Overall, I think the authors are careful not to over-interpret their results. My primary concern lies in the metric of species richness used, and whether it is misleading to refer to it as 'species richness' per se. Given the coarse scale of the measurement, it is more a metric of regional richness, or the richness of the species pool. The authors certainly don't try to hide this fact, but I think that it needs to be highlighted throughout the ms to avoid having the results be misinterpreted by a casual reader. Some specific areas where the language should be more precise include lines 51-52, line 127 (simply the addition of "regional coral richness" would work here), line 143, line 153.

Additional comments

The authors present an interesting analysis of the relationship between regional species richness and the response to disturbance in coral communities. I think that with additional clarification of the analyses and results this study will be a solid contribution to the ecological literature.

---

## Round 0.2 · Minor Revisions

We have had some problems getting a 2nd review of this manuscript, generally I believe because the reviewers were content with the responses of the authors to their previous reviews. We could push again for more reviews but, given the comprehensive response of the author to the initial reviews and the positive response of the reviewer that did respond, I would rather see this paper published (and the students who participated in its formulation rewarded for their hard work!)

·

Basic reporting

Cardinale et al. 2012 is a relevant reference to include in the introduction, particularly in lines 35-36.

Experimental design

The distinction between a 10% cutoff based on absolute decline (Graham et al. 2011) vs. relative coral decline (this study) is not completely clear (lines 98-101). Aren't both relative, in that they are based on pre-disturbance cover?

In the results section, it is easy to miss the 'recovery' results, since they only comprise a sentence at the end of a lengthy discussion of 'resistance' results. It may be helpful to set this result (lines 143-145) in a separate paragraph to emphasize that it comes from a separate analysis.

Validity of the findings

Re: my previous concerns over what to call the metric of species richness, I think that "predicted coral richness" is a good solution.

I found the discussion (lines 204-210) regarding the potential artifact of acroporid decline in the Caribbean hard to follow. Please clarify how the decline in acroporids before the studies included in this analysis were conducted is expected to affect coral cover, coral richness, and coral susceptibility to disturbance, particularly for a reader not familiar with coral ecology.

I think the points raised in the final paragraph regarding the scale of the richness estimates and their potential affect on the perceived relationship between diversity and stability should be expanded by a few sentences. Figure 2 really drives home the fact that there is relatively little variation in estimated richness within geographic regions, and thus the detection of a richness-recovery relationship depends primarily on how these factors are related across regions (i.e., richness and region are difficult to tease apart).

Additional comments

The authors have done a commendable job of addressing reviewer concerns, and the manuscript is much-improved. My additional suggestions (above) involve emphasizing or clarifying sections of the revision, and do not involve major changes. Nice work!

---

## Round 0.3 · accepted · Accept

Nice paper and an exemplary illustration of the potential for links between research and teaching.